# Intranasal challenge with *B. pertussis* leads to more severe disease manifestations in mice than aerosol challenge

**Kelly L. Weaver**[☯]**, Graham J. Bitzer**[☯]**, M. Allison Wolf, Gage M. Pyles, Megan A. DeJong, Spencer R. Dublin, Annalisa B. Huckaby, Maria de la Paz Gutierrez, Jesse M. Hall, Ting Y. Wong, Matthew Warden, Jonathan E. Petty, William T. Witt, Casey Cunningham, Emel Sen-Kilic, F. Heath Damron\*, Mariette Barbier**[ID]**\***

Vaccine Development Center in the Department of Microbiology, Immunology, and Cell Biology at West Virginia University, Morgantown, WV, United States of America

☯ These authors contributed equally to this work.
\* mabarbier@hsc.wvu.edu (MB); fdamron@hsc.wvu.edu (FHD)

## Abstract

The murine *Bordetella pertussis* challenge model has been utilized in preclinical research for decades. Currently, inconsistent methodologies are employed by researchers across the globe, making it difficult to compare findings. The objective of this work was to utilize the CD-1 mouse model with two routes of challenge, intranasal and aerosol administration of *B. pertussis*, to understand the differences in disease manifestation elicited via each route. We observed that both routes of *B. pertussis* challenge result in dose-dependent colonization of the respiratory tract, but overall, intranasal challenge led to higher bacterial burden in the nasal lavage, trachea, and lung. Furthermore, high dose intranasal challenge results in induction of leukocytosis and pro-inflammatory cytokine responses compared to aerosol challenge. These data highlight crucial differences in *B. pertussis* challenge routes that should be considered during experimental design.

## Introduction

*Bordetella pertussis* (*B. pertussis*) is a Gram-negative bacterium responsible for the human respiratory disease known as pertussis, or whooping cough. *B. pertussis* is an air-borne pathogen that is transmitted through exhaled respiratory droplets. Although *B. pertussis* is exclusively a human pathogen, animal models such as mice are commonly utilized to study *B. pertussis* pathogenesis and to direct vaccine development. Delivery of *B. pertussis* to the airways can lead to colonization of both the upper and the lower respiratory tracts. In animal models, the route of administration (intranasal or aerosol), dose, and the volume of inoculum used can effect colonization [1–3]. There are various pre-clinical animal models used to study pertussis, including mice, rats, pigs, and green olive baboons [4–8]. Pertussis disease manifestation can differ between models and can vary depending on the species, age, infectious route, and dose [7–10]. While all these models are useful, none of them perfectly recapitulate human pertussis infection. Mice are one of the most broadly used models of pertussis in the field. Both neonatal and adult

**Funding:** 1. FHD was supported by the National Institutes of Health grants 1R01AI137155-01A1

and 1R01AI153250-01A1. https://www.nih.gov/. The sponsor did not play any role in the study design, data collection and analysis, decision to publish, or preparation of the manuscript. 2. MB was supported by the National Institutes of Health grant 1R01AI14167101A1. https://www.nih.gov/. The sponsor did not play any role in the study design, data collection and analysis, decision to publish, or preparation of the manuscript. 3. K.L.W. received funding from the Cell and Molecular Biology and Biomedical Engineering Training Program funded by the National Institutes of Health NIGMS grant T32 GM133369 awarded to Schaller (PI). https://www.nih.gov/. The sponsor did not play any role in the study design, data collection and analysis, decision to publish, or preparation of the manuscript. 4. KLW was supported by the NASA West Virginia Space Grant Consortium Graduate Research Fellowship Program, Grant #80NSSC20M0055 (2021-2022).https://www. wvspacegrant.org/. The sponsor did not play any role in the study design, data collection and analysis, decision to publish, or preparation of the manuscript. 5. FHD and MB were supported by The WVU Vaccine Development Center, which in turn was supported by a Research Challenge Grant No. HEPC.dsr.18.6 from the Division of Science and Research, WV Higher Education Policy Commission. https://www.wvhepc.edu/. The sponsor did not play any role in the study design, data collection and analysis, decision to publish, or preparation of the manuscript." 6. There was no additional external funding received for this study.

**Competing interests:** The authors have declared that no competing interests exist.

mice have contributed to the advancement of knowledge of *B. pertussis* pathogenesis [11–13]. Additionally, mice have served a critical role in pre-clinical vaccine development and vaccine lot efficacy validation [14–17]. In the literature, *B. pertussis*, methodologies are often inconsistent (age, sex, mouse strain, method of infection, etc.) making comparison across studies difficult. To best contextualize the data obtained in each study it is important to understand how the challenge method and infectious dose of *B. pertussis* can alter disease manifestation in mice.

Intranasal (IN) challenge has been utilized in pertussis animal studies dating back to the 1930's [18]. To perform this technique, a pipette is used to administer a bacterial suspension directly to the external nares of mice. This procedure is performed under anesthesia to minimize ingestion of the bacteria orally [7, 19]. Depending on the volume used, the infection can be targeted to the upper respiratory tract (low inoculation volume) or both the upper and the lower respiratory tract (higher inoculation volumes) [1]. This model recapitulates human disease, as the bacteria colonize the respiratory tract and induce leukocytosis, a hallmark disease manifestation. IN challenge results in high bacterial burden within the respiratory tract, specifically within the lungs [20, 21]. This model differs from what is often observed in human patients suffering from pertussis, in which bacteria are most often associated with the upper respiratory tract [22, 23]. Therefore, while the use of intranasal administration is widespread and has provided important insights for the field, it is not completely representative of natural infection.

Aerosol challenge, in which bacteria are nebulized to mice housed in a chamber, is more closely representative of natural infection and has been used to study pertussis in animal models since the 1980's [7, 13, 24–29]. In this technique, mice inhale aerosolized bacteria as they breathe within a sealed chamber. Similar to the intranasal model, aerosol challenge permits bacterial colonization of the respiratory tract and induces leukocytosis [9, 25, 30]. In this work, we compared two routes of murine respiratory challenge, intranasal and aerosol, and the effect of *B. pertussis* challenge doses (ranging from $10^9$ CFU/mL to $2 \times 10^2$ total CFUs) in mice. The objective was to determine the impact of route and dose have on disease outcome. We hypothesized that aerosol challenge would better recapitulate human disease at it is a more physiologically relevant route of infection compared to intranasal challenge. We challenged outbred 10-week-old, female CD-1 mice with a range of *B. pertussis* doses. Mice were euthanized at 1 hour, 3 days, or 7 days post-challenge to quantify: 1) bacterial burden in the respiratory tract, 2) white blood cell counts (WBC), and 3) pro-inflammatory cytokine levels.

First, we observed that both intranasal and aerosol administration of *B. pertussis* led to dose-dependent colonization of the respiratory tract in CD-1 mice. However, compared to aerosolization, intranasal administration of a high *B. pertussis* dose led to greater persistence in the nares and trachea. Following infection, the distribution of bacterial burden between the upper, middle, and lower respiratory tracts correlated with the dose of *B. pertussis* administered rather than the route of administration. Leukocytosis, a marker of severe disease manifestation, was induced by high doses of *B. pertussis* administered via intranasal inoculation. Finally, the data suggests that a high intranasal challenge dose significantly increases pro-inflammatory cytokine levels in the lung supernatant. Taken together, these findings demonstrate that there are critical differences in disease manifestation based on both the route and dose of *B. pertussis* administered that should be considered during the experimental design of challenge studies utilizing *B. pertussis*.

## Methods

### *B. pertussis* strains and growth conditions

*Bordetella pertussis* strain UT25Sm1 was kindly provided by Dr. Sandra Armstrong (University of Minnesota). Isolate D420 was provided by Michael Weigand (Centers for Disease

Control). Both UT25 and D420 have been genome sequenced UT25Sm1 NCBI Reference Sequence is NZ_CP015771.1 and D420 is NZ_LN849008. UT25 was originally isolated in 1977 from a child diagnosed with pertussis [31, 32]. D420 was isolated in 2002 from an infant with severe respiratory disease in Texas [5, 33]. *B. pertussis* was grown on Difco™ Bordet Gengou (BG) agar (VWR™, Cat. #90003–414) supplemented with 15% defibrillated sheep blood (Hemostat Laboratories, Cat. #DSB500) and streptomycin 100 µg/mL (Gibco™, Cat. #11860038) for UT25 or cephalexin 40 µg/mL (Sigma-Aldrich, Cat. #C4895) for D420 at 36°C for 48 hours. For each bacterial challenge, the number of viable bacteria, the Bvg+ status, and the phenotype (hemolytic and characteristic colony morphology) was confirmed to ensure consistency between each challenge. Bacteria were then collected using polyester swabs (Fisher Scientific, Cat. #22-029-574) and resuspended in Stainer Scholte media (SSM) supplemented with L-proline and SSM supplement [34]. SSM liquid culture was incubated for 24 hours at 36°C with constant shaking at 180 rpm until reaching mid-log phase $OD_{600nm}$ 0.5 with 1 cm path width (Beckman Coulter™ DU 530 UV Vis spectrophotometer). *B. pertussis* culture was diluted in supplemented SSM to $OD_{600nm}$ = 0.24–0.245 (equivalent to $10^9$ CFU/mL) and serially diluted 1:10 to obtain doses $10^8$ CFU/mL, $10^7$ CFU/mL, $10^6$ CFU/mL, $10^5$ CFU/mL, $10^4$ CFU/mL, $10^3$ CFU/mL, and $10^2$ CFU/mL to be used for challenge. To aerosol challenge, 20 mL total of each culture was distributed amongst four nebulizers in 5 mL aliquots.

## *B. pertussis* challenge

In these studies, 10-week-old outbred CD-1 (Charles River), female mice were utilized (n = 5 per group). For intranasal challenge of animals, mice were anesthetized by intraperitoneal injection (IP) of ketamine (77 mg/kg) (Patterson Veterinary, Cat. #07-803-6637) and xylazine (7.7 mg/kg) (Patterson Veterinary, Cat. #07-808-1939) in sterile 0.9% NaCl (Baxter, Cat. #2F7124) and challenged intranasally with live *B. pertussis* (10 µL per nostril). For aerosol challenge mice were acclimated for 5 minutes in the DSI Buxco® FinePointe™ Mass dosing chamber (36.6 liters; internal chamber dimensions: 17.5"x11.75"). Mice were exposed in groups with up to 15 mice per group. The DSI mass dosing controller was set to 100% duty cycle percentage keeping the four nebulizers (5 mL culture per nebulizer) (4 to 6 µm Aerogen®, Cat# AG AL-1000) on fulltime and the aerosolization period was set to 10 minutes with fresh air flow rate set to 2 liters per minute (LPM). Upon completion of aerosolization the controller continued to supply fresh air at a flow rate of 5 LPM for 5 minutes before the mice were removed from the chamber. Mice were euthanized by IP injection of Euthasol (390 mg pentobarbital/kg) (Patterson Veterinary, Cat. #07-805-9296) in sterile 0.9% w/v NaCl at 1 hour, 1 day, 3 days, or 7 days post-challenge. Additional details regarding the two challenge methodologies utilized here have been published elsewhere [35]. Challenge studies were performed with both UT25Sm1 and D420 strains. Figs 2 to 6 and S1 to S4 Figs represent data from challenge studies conducted with UT25Sm1, while S5 and S6 Figs represent data from challenge studies conducted with D420.

## Complete blood count

Whole blood was collected via cardiac puncture into BD Microtainer® blood collection tubes with $K_2EDTA$ (BD, Cat#365974) to prevent coagulation. Whole blood parameters were analyzed using the IDEXX ProCyte Dx Hematology Analyzer.

## Quantification of bacterial burden

Lung homogenate, trachea homogenate, and nasal lavage fluid (nasal wash) were collected post-mortem to enumerate bacterial burden per tissue. The nasal cavity was flushed with 1 mL

sterile PBS to collect nasal lavage fluid. The lung and trachea were homogenized separately in 1 mL sterile PBS using a Polytron PT 2500 E homogenizer (Kinematica). Samples were serially diluted in ten-fold dilutions in PBS and plated on BG agar to quantify viable bacterial burden. Plates were incubated at 36˚C for 48–72 hours to determine colony forming units (CFUs) per mL.

### Cytokine analysis

Cytokines (IL-6, TNFα, IFN-ɣ, IL-17a, IL-10, GM-CSF) and chemokine CXCL13 in lung supernatant were measured using the Luminex® Discovery Assay mouse premixed multi-analyte kit (R&D Systems, Cat#LXSAMSM). Samples were diluted 1:2 in the provided buffer and the assay was performed according to the manufacturer's instructions. Samples were analyzed on a MAGPIX instrument (Luminex Corporation) using seven standards (in duplicate) for each plate analyzed. A total of 50 μL of sample was used for the assay and a bead count cutoff of 35 was used to invalidate any sample results below that cutoff.

### Graphing, data, and statistical analysis

All animal experiments were performed with n = 5 mice/group. Data was graphed and statistical analysis was performed using GraphPad Prism version 9. A one-way ANOVA with a Tukey's multiple comparison test was used when comparing three or more groups of parametric data while a Kruskal-Wallis test with a Dunnet's post-hoc test was used for non-parametric data. Principle component analysis was done using ClustVis web tool [36].

### Animal care and use

All mouse experiments were approved by the West Virginia University Institutional Animal Care and Use Committees (WVU-ACUC protocol 1901021039 and 1602000797) and completed in strict accordance of the National Institutes of Health Guide for the care and use of laboratory animals. All work was done using universal precautions at BSL2 under the IBC protocol # 17-11-01.

## Results

### Intranasal and aerosol administration of *B. pertussis* leads to dose-dependent colonization of the respiratory tract in mice

The objective of this work was to compare intranasal and aerosol respiratory challenge using a range of doses of *B. pertussis*. In order to compare the two routes of respiratory challenge, ten-week-old female CD-1 mice (n = 5) were challenged with *B. pertussis* and processed 1 hour, 3 days, or 7 days post-challenge (**Fig 1**). We first determined the efficiency of each route of infection by quantifying the bacterial burden in the respiratory tract at 1-hour post-challenge. To do this, the lung, trachea, and nasal lavage fluid were processed to quantify the number of bacteria in each sample. We observed that administration of *B. pertussis* by either methodology led to the detection of bacteria in both the upper and the lower respiratory tracts (**Fig 2A–2C**). At 1 hour post-challenge, there was a linear correlation between the dose administered and the number of bacteria recovered in the lung (**Fig 2D and 2E**). Overall, we observed that aerosol-challenged mice had lower bacterial burden in the respiratory tract compared to mice that were intranasally challenged (**Fig 2A–2C**). At 1-hour post-challenge, the bacterial burden of mice infected intranasally with $2x10^7$ CFUs was 32 times higher than that observed with the $10^9$ CFU/mL aerosol challenge. Furthermore, an aerosolization dose of $10^9$ CFU/mL was comparable to an intranasal challenge with $2x10^6$ CFUs at 1-hour post-challenge (**Fig 2A–2C**). To

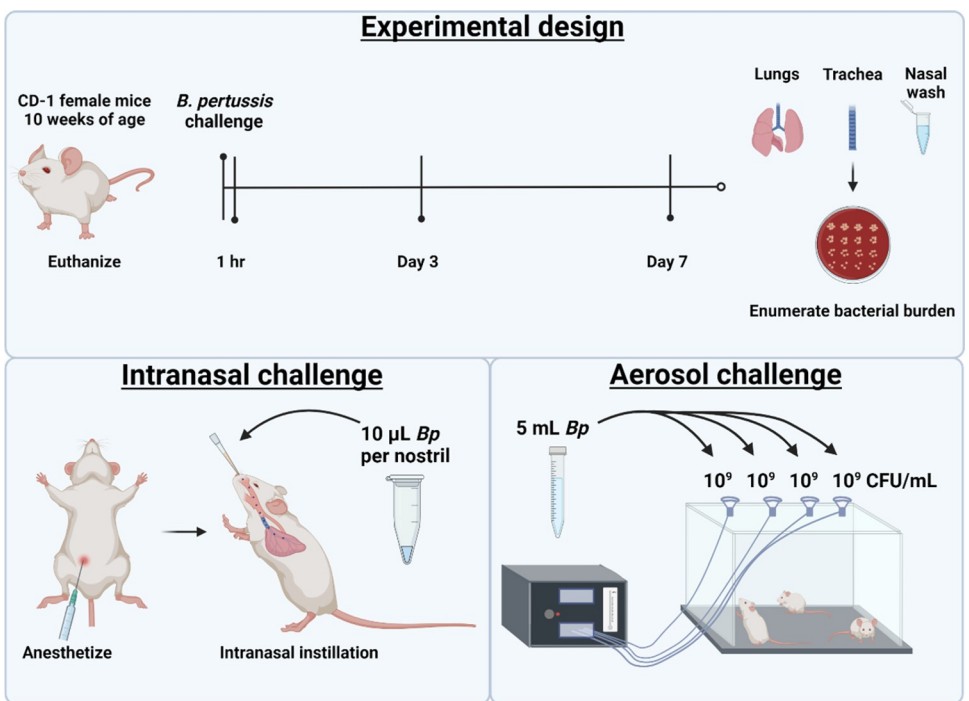

**Fig 1. Experimental design and routes used for challenge.** In this study 10-week-old female CD-1 mice were challenged with *B. pertussis* and euthanized at 1 hour, 3 days, or 7 days post-challenge. The lungs, trachea, and nasal lavage fluid were collected to enumerate bacterial burden at each time point. Intranasal administration of *B. pertussis* was compared to aerosol challenge using a range of challenge doses.

compare disease outcome with each infection route in the remainder of this work, we selected doses with equivalent bacterial loads at 1-hour post-challenge. Infectious doses of $10^6$ CFU/mL aerosol and $2\times10^2$ total CFUs IN both led to the recovery of approximately 10 colony forming units in each tissue 1-hour post-challenge and were considered as "low doses". Infectious doses of $10^9$ CFU/mL aerosol and $2\times10^6$ total CFUs IN both led to the recovery of approximately $10^5$ CFUs in the lung, and $10^4$ CFUs in the trachea and nasal washes after 1 hour and were considered "high doses". These doses were compared throughout the remainder of this study to assess disease outcomes in animals that received comparable initial infectious doses.

**Intranasal administration of a high dose of *B. pertussis* leads to greater persistence of *B. pertussis* over time compared to aerosol administration.** Bacterial colonization in mice challenged with a high or low infectious doses via the aerosolization or intranasal methods was quantified at 1 hour, 3 days, and 7 days (**S2 Fig**). Viable bacteria present in nasal lavage, trachea, and lung were enumerated using serial dilutions and plating. We observed that administration of low doses of bacteria using either intranasal or aerosol routes led to the detection of similar bacterial loads in the nasal wash, trachea, and lung over time (**Fig 3**). While the bacterial loads remained constant in the nasal wash and trachea, we observed an increase in bacterial burden in the lung at day 7 compared to 1-hour post-challenge ($p = 0.9343$ for low aerosol; $p = 0.5636$ for low IN) (**Fig 3**). We also observed that an infectious dose in mice as low as 100 bacteria administered IN was sufficient for the persistence and colonization of *B. pertussis* in the respiratory tract.

Interestingly, despite a similar deposit of bacteria at 1-hour, the progression of bacterial burden over time was very distinct in mice that received the high infectious dose. Animals challenged with a high dose via aerosol maintained a relatively constant bacterial burden in

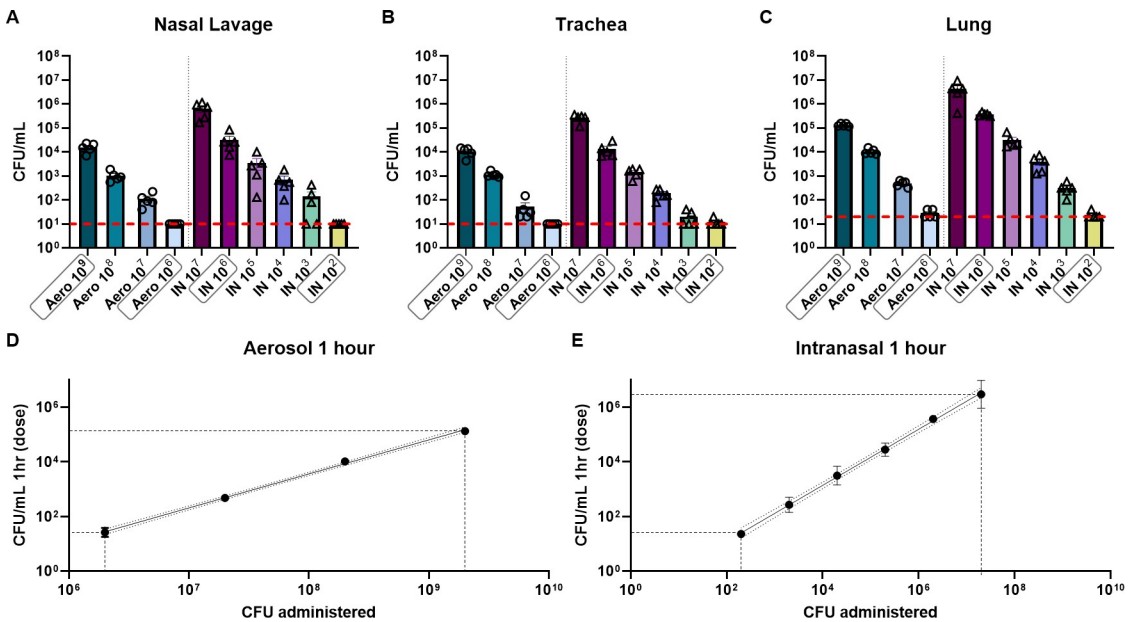

**Fig 2. Intranasal and aerosol administration of *B. pertussis* (UT25Sm1) lead to dose-dependent colonization of the respiratory tract in mice.** Mice were challenged utilizing an aerosol chamber (20 mL total) with doses of *B. pertussis* from $10^9$ CFU/mL to $10^6$ CFU/mL, or by intranasal administration (20 µL total) of *B. pertussis* with doses from $10^7$ CFU to $10^2$ CFU. Bacterial burden was quantified one-hour post-challenge in the **A)** nasal lavage fluid, **B)** trachea, and **C)** lung. Error bars are mean ± SEM (n = 5 per group). A simple linear regression was performed on Y = Log(Y) transformed CFU data to compare the dose administered by **D)** aerosol challenge **E)** to the number of bacteria recovered from the lung. High and low doses for each challenge route are highlighted with a gray box. This figure was created with BioRender.com and is published under a CC BY license, with permission from Biorender, original copyright 2023.

both the upper and lower respiratory tracts (**Fig 3**). In contrast, mice challenged with a high dose IN had significantly higher bacterial burden in the lung and nasal wash at day 3 (**Fig 3C**) compared to mice infected via the aerosol route with a high infectious dose. In these mice, the levels of viable bacteria recovered from the lung decreased at day 7, but continued to increase in the trachea (**Fig 3B**). Overall, mice infected intranasally with the high dose of *B. pertussis* maintained greater levels of viable bacteria in the airways 7 days post-infection compared to

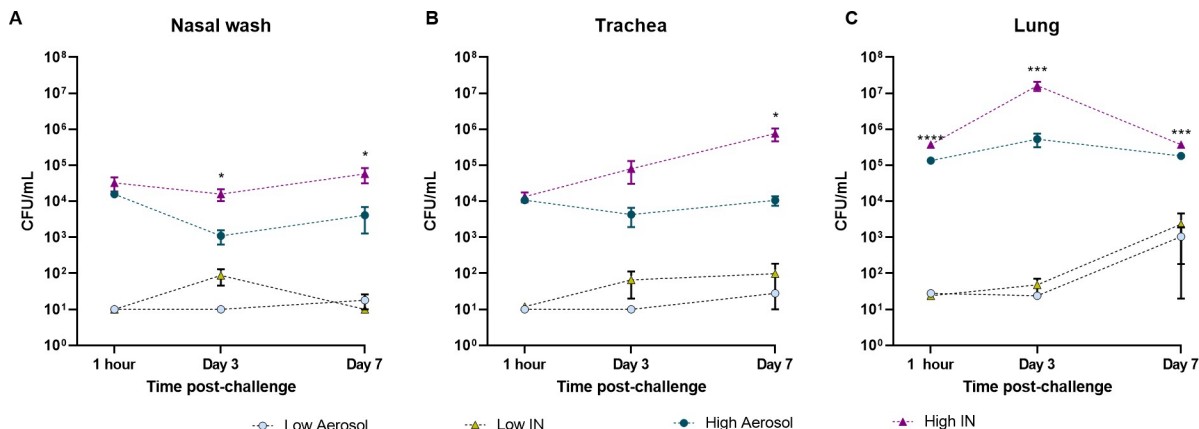

**Fig 3. Intranasal administration of high doses of *B. pertussis* (UT25Sm1) leads to increased colonization of the respiratory tract compared to aerosol challenge.** Bacterial burden in the **A)** nasal lavage fluid, **B)** trachea, and **C)** lung. The *p*-values were calculated using ANOVA followed by a Tukey's multiple-comparison test, *$p < 0.05$, ***$p < 0.001$, and ****$p < 0.0001$. Error bars are mean ± SEM values.

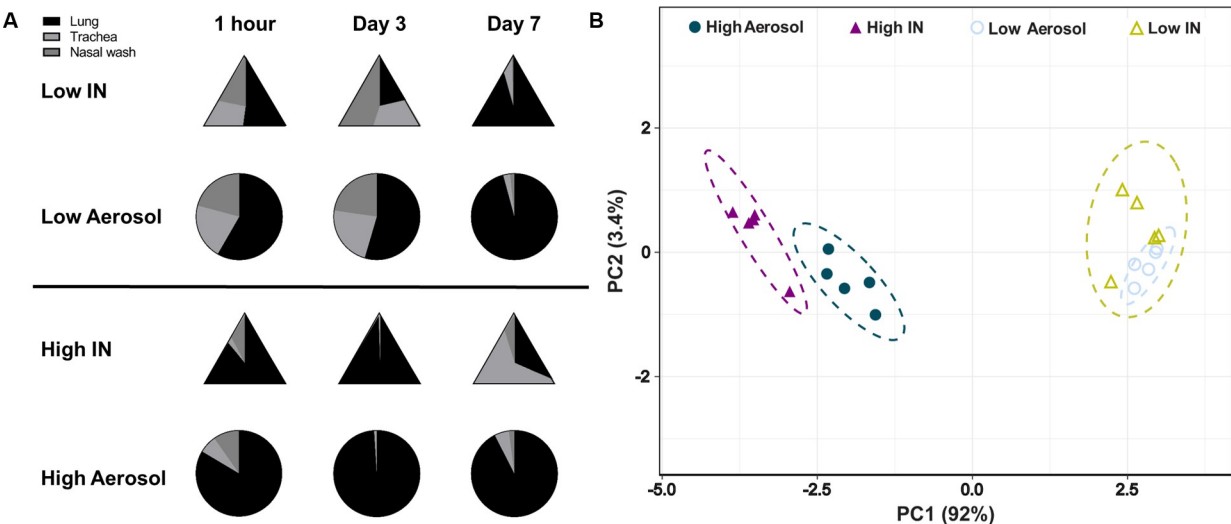

**Fig 4. The distribution of bacteria in the nasal wash, lung, and trachea correlates with the challenge dose rather than the route of administration. A)** Visual representation of the bacterial burden in the nasal lavage fluid, trachea, and lung at each time point. **B)** Principle component analysis of the bacterial burden for each challenge group.

mice that received a high dose of *B. pertussis* via aerosol. The kinetics of the progression of bacterial burden over time were very different between each model and each dose, emphasizing the importance to further evaluate the role of these parameters during *B. pertussis* infection.

**Distribution of bacterial burden following infection correlates with bacterial dose rather than the method of administration.** Infections with *B. pertussis* in humans are primarily associated with colonization of the upper respiratory tract and are more rarely associated with pneumonia [37–39]. To determine if infectious dose and routes had an impact on the distribution of the number of viable bacteria in the upper, middle, and lower respiratory tract of mice, we compared the proportion of viable bacteria from each tissue over time (**Fig 4A**). First, we observed that at 1-hour post-challenge, viable bacteria were relatively evenly distributed in the respiratory tracts of mice administered with a low dose, either IN or via aerosol (approx. 50% in the lung, 25% in the trachea, and 25% in the nasal wash). Interestingly, mice administered a high infectious dose of *B. pertussis* had a much higher number of viable bacteria recovered from the lung compared to the trachea or the nasal lavage fluid 1-hour post-challenge, regardless of the method used for infection. When comparing data over time, bacterial distribution at day 3 was similar to 1-hour post-challenge for all groups. At day 7 post-challenge, most of the viable bacteria were recovered from the lower respiratory tract, for mice receiving low doses (IN and aerosol) and the high aerosol dose. Surprisingly, mice that had received a high dose intranasally had higher bacterial burden in the trachea at day 7 compared to lung and nasal wash. Principal component analysis of the distribution of bacterial burden over time highlighted that overall, the distribution of bacteria between the nares, trachea, and lung was predominantly associated with the dose administered and not with the route (**Fig 4B**).

## Leukocytosis is associated with a high intranasal challenge dose

While *B. pertussis* infection in humans is known for the hallmark paroxysmal whooping cough, it is also responsible for a significant increase in circulating white blood cells such as neutrophils [40]. Leukocytosis is caused by pertussis toxin and is a marker of severe disease in

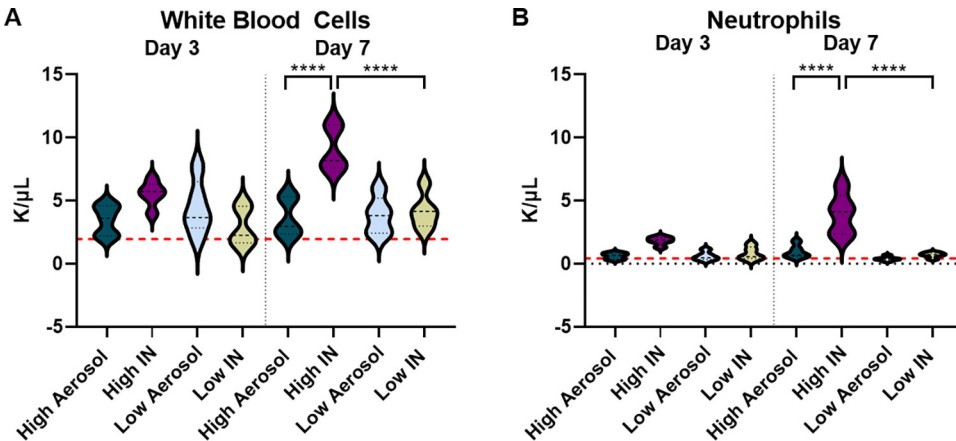

**Fig 5. White blood cell and neutrophil counts in whole blood following infection (*B. pertussis* UT25Sm1). A)**
White blood cells and **B)** neutrophils were measured in the whole blood a days 3 and 7 post-challenge. The p-values were calculated using ANOVA followed by a Tukey's multiple-comparison test, **** $p < 0.0001$ (n = 5 per group). Error bars are mean ± SEM values.

infants [30]. As such, white blood cell counts and neutrophil counts were measured at days 3 and 7 post-challenge from whole blood. We observed that intranasal administration of high doses of *B. pertussis* was associated with a significant increase in both white blood cells (**Fig 5A**), and neutrophils (**Fig 5B**) at day 7 post-challenge compared to mice that received a high dose aerosol challenge. There was also a significant increase in white blood cells and neutrophils in the high dose intranasal group compared to the low dose intranasal group. Significant increases in white blood cells and neutrophils were not observed with intranasal administration of low doses of *B. pertussis*, nor with aerosol challenge (**S3 Fig**) at either time point. The data suggests that to evaluate the effects of *B. pertussis* challenge in the context of leukocytosis, the intranasal model may be more advantageous compared to the aerosol challenge model for the doses and time points utilized in this study.

## High dose intranasal challenge leads to a significant increase in cytokine levels measured in the lung supernatant of challenged mice

Cytokines play an important role in inflammation and immunity and as such, can contribute to both the severity and the alleviation of disease [41]. During *B. pertussis* infection, there is an increase in inflammatory responses and cytokines such as interferon gamma (IFN-γ), tumor necrosis factor alpha (TNF-α), and Interleukin (IL-6) [20, 42, 43]. Numerous additional other cytokines are expressed, including those involved in the polarization of adaptive T cell responses such as IL-17. To determine the impact of both bacterial dose and challenge routes on the cytokine response, we measured granulocyte-macrophage colony-stimulating factor (GM-CSF), TNF-α, chemokine ligand 13 (CXCL13), IL-6, Interleukin-10 (IL-10), Interleukin-17 (IL-17/IL17-A), and IFN-γ in the lung supernatant following *B. pertussis* challenge. Analyses were performed at 1 hour, 3 days, and 7 days post-challenge to gain insights into the development of cytokine expression over time.

IL-6 is a cytokine that plays an important role in both innate and adaptive immune responses including T cell proliferation, B cell differentiation, and the induction of chemokines expression. IL-6 is known to play a significant role in mounting an immune response to infection with *B. pertussis* and vaccination against pertussis [20, 42]. C57BL/6, CD-1, and BALB/C mice have an increase in IL-6 production post-infection with *B. pertussis* [20, 21, 42, 44]. Harvill *et al* also

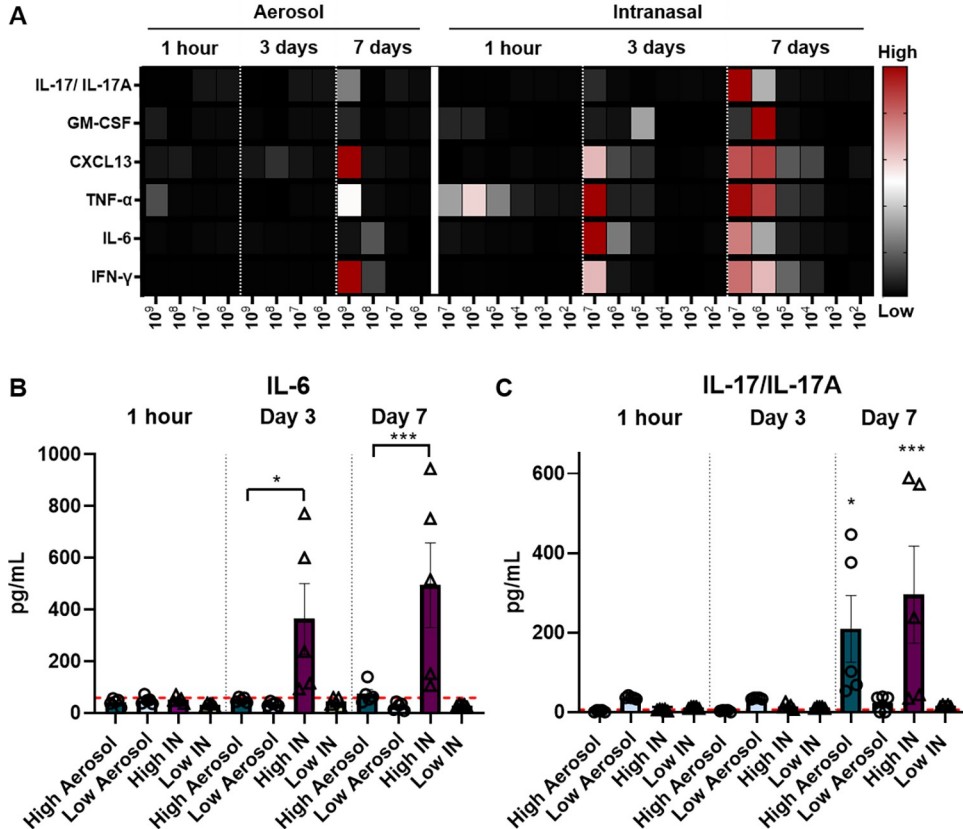

**Fig 6. High dose intranasal challenge (UT25Sm1) leads to a significant increase in cytokine responses compared to aerosol challenge.** The dose is shown on the x-axis. Values are in pg/mL on the y-axis. A) Heat map of cytokines measured in the lung supernatant at 1 hour, day 3, and day 7 post-challenge. A) IL-6 and B) IL-17/IL-17A represented as pg/mL. The p-values were calculated using ANOVA followed by a Tukey's multiple-comparison test, * $p < 0.05$ and *** $p < 0.001$ (n = 5 per group). Stars represent comparison to baseline unless otherwise indicated. Error bars are mean ± SEM values.

showed that there is a delay in clearance of *B. pertussis* in IL-6 deficient mice which is attributed to disruption of the adaptive immune response [44]. In this work IL-6 induction was not detectable until day 3 post-IN-challenge. Furthermore, a significant increase in IL-6 was only observed in the lungs of mice challenged with a high intranasal dose compared to mice challenged with a high aerosolized dose (**Fig 6A**) at day 3 and 7. Low dose challenge, regardless of the mode of infection, was not sufficient to induce IL-6 release at the time points evaluated.

IL-6 is associated with the induction of Th17 responses and the secretion of IL-17 in response to *B. pertussis* infection [20, 45]. Here, IL-17/IL-17A levels were measured in the lung supernatant from 1-hour, day 3, and day 7 post-challenge with *B. pertussis*. Unlike IL-6, IL-17 was not detectable in the lung of infected animals until day 7. We observed that by day 7 post-challenge, there was a significant increase in IL-17/IL-17A in groups that were administered a high dose of *B. pertussis* by either aerosol or intranasal instillation compared to naïve mice (**Fig 6A**). Again, no induction of IL-17 release was observed in mice that received low doses of *B. pertussis*.

GM-CSF, CXCL13, TNF-α, and IFN-ɣ were also measured in the lung supernatant following *B. pertussis* challenge (**Fig 6**). GM-CSF is a cytokine that plays roles in both immune responses and inflammation [46, 47]. CXCL13 increases in response to infection and is classically

associated with germinal center formation during adaptive immune responses [48, 49]. IFN-ɣ is crucial for clearance of *B. pertussis*, and depletion of IFN-ɣ leads to an increase in bacterial burden in IFN-ɣ depleted mice compared to controls [43]. Overall, *B. pertussis* challenge was associated with a dose-dependent increase in cytokines responses (**Fig 6A, S4 Fig**). Low doses of *B. pertussis* administered intranasally did not induce a measurable increase in the expression of these cytokines at any of the time points evaluated. Similar results were observed for low doses of *B. pertussis* administered via the aerosol route. Cytokine expression was measurable at days 3 and 7 post-challenge in mice challenged IN, while aerosol challenge did not lead to significant increases in cytokine responses until day 7 post-challenge with a high dose of bacteria.

In addition to cytokine measurements, wet lung weights were recorded as a crude measurement of overall inflammation and edema in response to challenge. Although none of the comparisons performed between aerosol and intranasal challenge were statistically significant, similar trends to those observed for cytokines were recorded: wet lung weights of mice challenged by aerosol remained low over the course of 7 days, while the overall wet lung weight of mice challenged IN was more elevated, in particular 3 days post-challenge (**S1 Fig**).

## Differences observed between infection routes are independent of the bacterial challenge strain

Altogether, these data highlight the differences between intranasal administration of *B. pertussis* compared to aerosolization, and how the dosage used impacts disease manifestation in CD-1 mice. While the data presented above was generated using the strain UT25, experiments were also performed with the strain D420, a recent clinical isolate used in baboon challenge studies [50]. We observed similar trends in bacterial burden (**S5 Fig**), leukocytosis, and cytokine expression (**S6 Fig**) in mice infected with D420 as those infected with UT25. This suggests that the differences observed between the infection routes and the doses are not strain-specific.

## Discussion

In this work we sought to understand the differences in intranasal and aerosol challenge in the CD-1 mouse model of pertussis. Altogether, we observed that intranasal and aerosol administration of *B. pertussis* leads to dose-dependent bacterial burden in the respiratory tract, yet there are differences observed in the disease profile between the routes. Intranasal challenge with a high dose of *B. pertussis* led to greater persistence of bacteria over time compared to aerosol challenge. Interestingly, the distribution of bacterial burden correlates with the bacterial dose rather than the challenge method. As pertussis is associated with leukocytosis in infants, we measured white blood cells and neutrophils in the whole blood. Only high dose intranasal challenge leads to a significant increase in leukocytosis. Next, we measured chemokines and cytokines in the lung supernatant of challenged mice to better understand the immune response induced by each challenge method. High dose intranasal challenge led to a significant increase in cytokine levels such as IL-6 and IL-17. These data highlight that both the dose and route of challenge are important to consider when designing murine challenge studies (**Fig 7**).

Murine challenge studies have been performed for decades to better understand *B. pertussis* infection and to inform vaccine development. Various methodologies have been used to challenge mice in these studies, including intracerebral challenge, intranasal instillation of bacteria to the nares, and aerosol administration via dosing chambers. While these methods have been informative for the field as a whole, the techniques and doses utilized vary from one study to the next making it difficult to compare findings. Prior to the development of the IN and aerosol challenge methodologies, the intracerebral protection assay, also known as the Kendrick

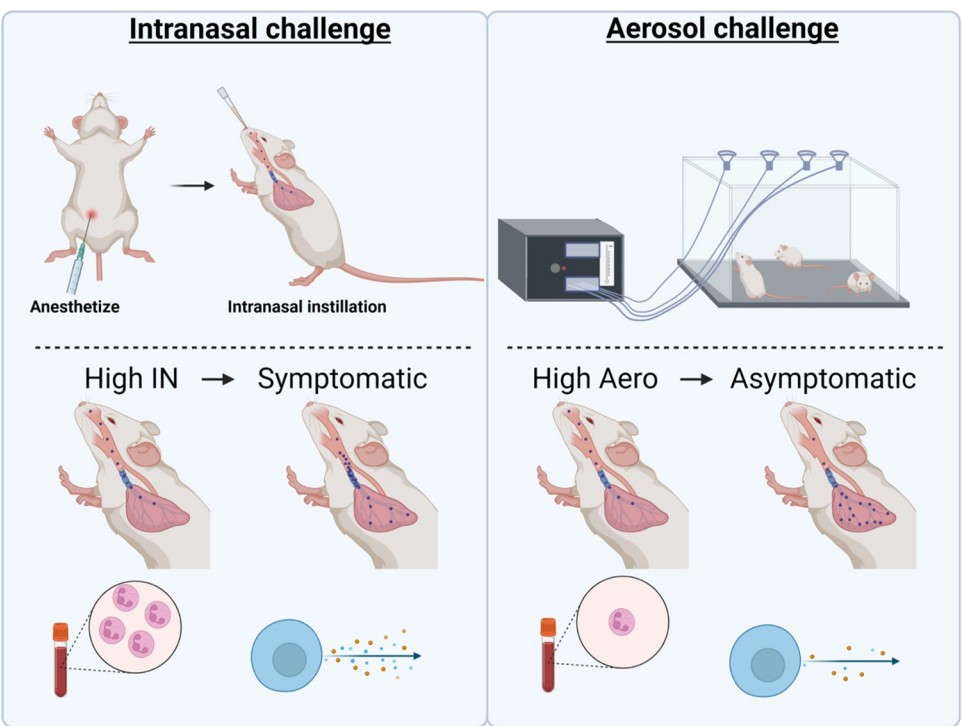

**Fig 7. Model comparing intranasal challenge and aerosol challenge at high doses.** Intranasal challenge is associated with an increase in bacterial burden over time. The distribution of bacteria following intranasal challenge is associated with the upper respiratory tract compared to aerosol challenge which is associated with the lower respiratory tract. Intranasal administration stimulates increases in leukocytosis and cytokine production compared to aerosol administration. This figure was created with BioRender.com and is published under a CC BY license, with permission from Biorender, original copyright 2023.

test, was developed to control the potency of whole cell pertussis vaccines. This model is advantageous as it correlates with protection in humans [14]. Classically, intracerebral challenge is performed two weeks post-vaccination to compare the efficacy of pertussis vaccine lots in comparison to reference control vaccines. Mice are monitored for lethal effects in the days after challenge. While this test is effective at determining vaccine potency and correlates with protection in humans, it has been criticized for its ethics, reproducibility, and for the fact that this is not a natural route of infection of *B. pertussis*.

In contrast to intracerebral challenge, there are two common modes of *B. pertussis* challenge that target the respiratory tract: intranasal administration of bacteria to the nares and aerosol administration using dosing chambers. Intranasal challenge models were implemented as early as 1937 to evaluate pertussis in mice. Intranasal challenge, otherwise known as droplet inoculation, can result in either lethal or sublethal infection depending on the dose administered. Lethal infection is associated with a decrease in body weight and temperature, spleen atrophy, and leukocytosis [7]. On the other hand, the sublethal model results in normal body weight and temperature, and lower levels of leukocytosis [7, 10]. In 1980, Sato *et al* developed an aerosol challenge model in effort to improve the reproducibility and uniformity of infection compared to the intranasal models at the time [25]. Their aerosol model was associated with an increase in lung weight and leukocytosis, and a decrease in weight gain in mice. This model utilized a high dose ($2\times10^9$ viable cells per mL) and effects were observed 14 days post-challenge. Similarly to the intranasal challenge model, younger mice had increased severity of infection and eventually succumbed to infection, while older mice had decreased susceptibility

and 100% survival. Over the years, both techniques have shown to be highly reproducible, reliable, and easy to implement, and have been extremely informative to understand *B. pertussis* pathogenesis and to inform vaccine development efforts.

In this study, we first compared the efficacy of intranasal and aerosol administration to infect mice with *B. pertussis*. Both intranasal and aerosol challenge methods allowed for the recovery of dose-dependent numbers of bacteria 1-hour post-challenge in the nasal wash (**Fig 2A**), trachea (**Fig 2B**), and in the lung homogenate (**Fig 2C**). Both methodologies had high reproducibility and a similar distribution of the bacteria into the respiratory tract over time (**Figs 3 and 4**). When administered at a high dose, mice infected intranasally maintained a higher bacterial burden in the nares, trachea and lung over time than mice infected via aerosol (**Fig 3**). Interestingly, the kinetics of bacterial persistence over time, as well as the distribution of bacterial burden in the respiratory tract, correlated with the dose administered rather than with the mode of administration (**Fig 4**). Finally, we observed that intranasal administration allowed for the administration of a higher starting load of bacteria than aerosol administration (**Fig 2**). While technically more challenging, it is conceivable that a higher dose could also be delivered using the aerosol route by increasing the volume of bacterial suspension aerosolized and the time of exposure. When comparing disease manifestations with both models, we observed that intranasal challenge overall resulted in stronger disease manifestation, including higher leukocytosis (**Fig 5**) and cytokine release (**Fig 6**) at days 3 and 7 post-challenge compared to aerosol challenge. Based on previous studies, the increased presence of neutrophils suggests that IL-17 is elevated and the data reported here support those findings [30, 45, 51]. In this work, we did not observe a significant increase in white blood cells (**Fig 5A**) or leukocytosis (**Fig 5B**) in aerosol challenged mice by day 7 post-challenge. This may be attributed, in part, to the higher bacterial burden detected in mice challenged with a high dose of *B. pertussis* intranasally compared to those challenged with a high dose by aerosol. Additional time points, such as day 14, could also prove valuable to compare leukocytosis levels to the original studies by Sato [25].

There are numerous factors that could influence the data reported in this study. Mouse and bacterial strains can have a major impact on the outcomes of infection models [7, 8, 52–54]. While we did not investigate the role of the mouse strain in this study, we performed intranasal and aerosol challenge studies using another strain of *B. pertussis*, D420. This strain has been used extensively for challenge studies in the green olive baboon model of pertussis [5, 50, 55]. In our studies, we observed that intranasal and aerosol infection with either UT25 or D420, at multiple doses, led to similar bacterial burden (**S4 Fig**), distribution in the airways, leukocytosis, and cytokine production in mice over the course of 7 days. In the future, these studies could be expanded to include additional bacterial and mouse strains. Other factors such as age and biological sex can also play a role during infection. For example, as observed in humans, young mice are more susceptible to infection compared to adult mice, and neonatal models of pertussis have proven to be valuable for studying whooping cough and informing vaccine development efforts [11, 12, 56]. The studies presented here were performed with 10-week-old female mice. Side-by-side comparison of males and females, and young versus adult mice could shed additional information regarding the differences between intranasal and aerosol challenge methods for *B. pertussis*.

In addition to the differences observed in bacterial colonization and disease manifestations, it is important to point out technical advantages and caveats of each methodology. One major feature of intranasal administration is that bacteria are deposited directly on the nostrils, avoiding exposure of the fur, skin, and eyes of the animals. In addition, intranasal delivery also allows for the control of the location of where bacteria are delivered. Small volumes (5–10 μL) are sufficient to deliver bacteria to the nasal cavity, while larger volumes (20–50 μL) facilitate

deposition of the bacteria in the lung. Variation of this methodology, such as pharyngeal aspiration, can provide additional flexibility by delivering bacteria only to the trachea and the lung by bypassing the nares. One caveat of the intranasal technique is that it requires anesthesia and the effects of anesthesia in this model are not well understood. Anesthesia using ketamine-xylazine, isofluorane, and urethane affects baseline breathing in mice [57]. Previous studies suggest that ketamine also induces immunosuppression which could affect the immune response to challenge [58]. Ketamine has immunomodulatory effects that can lead to significant suppression of cellular responses such as inhibition of macrophages and NK cells [58, 59]. This caveat is important to consider as intranasal administration of bacteria without anesthesia is not practical and is associated with animal welfare concerns. In contrast, aerosol challenge is advantageous in that it does not require anesthesia as bacteria are nebulized into the chamber and inhaled. This methodology also allows for the exposure of multiple animals at once, increasing throughput and reproducibility. Some caveats of aerosol administration include deposition of bacteria on the fur, skin, and eyes of the animals, and the need of well-calibrated and specialized equipment that requires thorough cleaning and decontamination after each use.

This study provides highly valuable information for the field by direct comparison of murine models of *B. pertussis* challenge. As new vaccine formulations are tested, it is important to maintain methodological consistency to compare efficacy and select candidates for baboon challenge studies and human clinical trials. In addition, while we know *B. pertussis* transmits via aerosol very efficiently among baboons, only anecdotal evidence has been presented to determine the minimal infectious dose for whooping cough [60]. In humans, it had been suspected that the infectious dose is relatively low, as household contact studies indicate high transmission rates [60–64]. Using both the intranasal and aerosol models, data from this study show that as few as 10–100 bacteria initially deposited in the airways by 1-hour are sufficient to lead to bacterial colonization and persistence for at least 7 days (**Fig 3**). To note, none of the mice infected with such small number of bacteria displayed changes in white blood cell counts or cytokine production. It is conceivable that these mice could be experiencing bacterial colonization similar to what is observed in asymptomatic human carriers capable of transmitting *B. pertussis* (**Fig 7**). The human challenge model will likely shed additional light on the etiology of *B. pertussis* including disease transmission and asymptomatic carriage, and help with bridging the gap between animal models and humans.

## Supporting information

**S1 Fig. Wet lung weight of intranasal and aerosol challenged mice at days 3 and 7 post-challenge.** Error bars are mean ± SEM values (n = 5 per group). ANOVA with Kruskal Wallis comparisons were performed for the high dose (Aerosol 109 vs IN 106) and no statistical differences were found between these groups at the timepoints depicted in this figure.
(TIF)

**S2 Fig. Bacterial burden (UT25Sm1) in the respiratory tract of intranasal and aerosol challenged mice.** Bacterial burden in the **A)** nasal lavage, **B)** trachea, and **C)** lung of mice at day 3 post-challenge. Bacterial burden in the **D)** nasal lavage, **E)** trachea, and **F)** lung of mice at day 7 post-challenge. Error bars are mean ± SEM values (n = 5 per group).
(TIF)

**S3 Fig. White blood cell and neutrophil counts measured in whole blood of challenged mice (UT25Sm1). A)** White blood cell counts and **B)** neutrophils measured at day 3 and day 7 post-challenge. Stars represent comparison to baseline. The *p*-values were calculated using

ANOVA followed by a Tukey's multiple-comparison test, $^*$ $p < 0.05$, $^{**}$ $p < 0.01$, and $^{***}$ $p < 0.001$ (n = 5 per group). Error bars are mean ± SEM values.
(TIF)

**S4 Fig. High dose intranasal challenge leads to a significant increase in cytokine responses compared to aerosol challenge (UT25Sm1). A)** IL-17/IL-17A **B)** GM-CSF, **C)** CXCL13, **D)** TNF-α, **E)** IL-6, and **F)** IFN-γ represented as pg/mL for all challenge doses. The p-values were calculated using ANOVA followed by a Tukey's multiple-comparison test, $^{**}$ $p < 0.01$, $^{***}$ $p < 0.001$, and $^{****}$ $p < 0.0001$ (n = 5 per group). Error bars are mean ± SEM values.
(TIF)

**S5 Fig. Bacterial burden in the respiratory tract of intranasal and aerosol challenged with *B. pertussis* strain D420.** Bacterial burden in the **A)** nasal lavage, **B)** trachea, and **C)** lung of mice at day 3 post-challenge. Bacterial burden in the **D)** nasal lavage, **E)** trachea, and **F)** lung of mice at day 7 post-challenge. Error bars are mean ± SEM values (n = 5 per group).
(TIF)

**S6 Fig. Measurement of cytokines in the lung of intranasal and aerosol challenged with *B. pertussis* strain D420 at 1h or 3 days post-challenge.** Data are represented as mean ± SEM values in pg/mL.
(TIF)

## Acknowledgments

The authors would like to acknowledge Michael Weigand at the Centers for Disease Control, Atlanta, GA for providing the strain D420. We would also like to acknowledge the OLAR facilities and staff at West Virginia University for the animal care and support of these studies.

## Author Contributions

**Conceptualization:** Kelly L. Weaver, Graham J. Bitzer, F. Heath Damron, Mariette Barbier.

**Data curation:** Kelly L. Weaver, Graham J. Bitzer, F. Heath Damron, Mariette Barbier.

**Formal analysis:** Kelly L. Weaver, Graham J. Bitzer, F. Heath Damron, Mariette Barbier.

**Funding acquisition:** Kelly L. Weaver, F. Heath Damron, Mariette Barbier.

**Investigation:** Kelly L. Weaver, Graham J. Bitzer, M. Allison Wolf, Gage M. Pyles, Megan A. DeJong, Spencer R. Dublin, Annalisa B. Huckaby, Maria de la Paz Gutierrez, Jesse M. Hall, Ting Y. Wong, Matthew Warden, Jonathan E. Petty, William T. Witt, Casey Cunningham, Emel Sen-Kilic.

**Methodology:** Kelly L. Weaver, Graham J. Bitzer, F. Heath Damron, Mariette Barbier.

**Project administration:** F. Heath Damron, Mariette Barbier.

**Resources:** F. Heath Damron, Mariette Barbier.

**Supervision:** F. Heath Damron, Mariette Barbier.

**Validation:** Graham J. Bitzer, F. Heath Damron, Mariette Barbier.

**Visualization:** Kelly L. Weaver, Graham J. Bitzer, F. Heath Damron, Mariette Barbier.

**Writing – original draft:** Kelly L. Weaver, Graham J. Bitzer, M. Allison Wolf, Gage M. Pyles, Megan A. DeJong, Spencer R. Dublin, Annalisa B. Huckaby, Maria de la Paz Gutierrez,

Jesse M. Hall, Ting Y. Wong, Matthew Warden, Jonathan E. Petty, William T. Witt, Casey Cunningham, Emel Sen-Kilic, F. Heath Damron, Mariette Barbier.

**Writing – review & editing:** Kelly L. Weaver, Graham J. Bitzer, M. Allison Wolf, Gage M. Pyles, Megan A. DeJong, Spencer R. Dublin, Annalisa B. Huckaby, Maria de la Paz Gutier-rez, Jesse M. Hall, Ting Y. Wong, Matthew Warden, Jonathan E. Petty, William T. Witt, Casey Cunningham, Emel Sen-Kilic, F. Heath Damron, Mariette Barbier.

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
