## [Decision Letter · Decision Letter 0]

27 Jan 2023

PONE-D-22-33512Intranasal challenge with B. pertussis leads to more severe disease manifestations in mice than aerosol challengePLOS ONE

Dear Dr. Barbier,

Thank you for submitting your manuscript to PLOS ONE. After careful consideration, we feel that it has merit but does not fully meet PLOS ONE’s publication criteria as it currently stands. Therefore, we invite you to submit a revised version of the manuscript that addresses the points raised during the review process.

1) The authors should make clear the strain used to generate the results linked to the figures and data;2) The control of the use of anesthesia can be of importance when comparing nasal versus aerosol challenges;3) Please, see the comments raised by all the reviewers.

We look forward to receiving your revised manuscript.

Kind regards,

Paulo Lee Ho, Ph.D.

Academic Editor

PLOS ONE

Journal Requirements:

"1. FHD was supported by the National Institutes of Health grants 1R01AI137155-01A1 and 1R01AI153250-01A1. 

https://www.nih.gov/. The sponsor did not play any role in the study design, data collection and analysis, decision to publish, or preparation of the manuscript.

2. MB was supported by the National Institutes of Health grant 1R01AI14167101A1. https://www.nih.gov/. The sponsor did not play any role in the study design, data collection and analysis, decision to publish, or preparation of the manuscript.

3. K.L.W. received funding from the Cell and Molecular Biology and Biomedical Engineering Training Program funded by the National Institutes of Health NIGMS grant T32 GM133369 awarded to Schaller (PI).  https://www.nih.gov/. The sponsor did not play any role in the study design, data collection and analysis, decision to publish, or preparation of the manuscript.

4. KLW was supported by the NASA West Virginia Space Grant Consortium Graduate Research Fellowship Program, Grant #80NSSC20M0055 (2021-2022). https://www.wvspacegrant.org/. The sponsor did not play any role in the study design, data collection and analysis, decision to publish, or preparation of the manuscript.

5. FHD and MB were supported by The WVU Vaccine Development Center, which in turn was supported by a Research Challenge Grant No. HEPC.dsr.18.6 from the Division of Science and Research, WV Higher Education Policy Commission. https://www.wvhepc.edu/. The sponsor did not play any role in the study design, data collection and analysis, decision to publish, or preparation of the manuscript."

3. We noted in your submission details that a portion of your manuscript may have been presented or published elsewhere. [A detailed version of the method section is currently under consideration at STAR Protocols. As small subset of the data is used in that manuscript as example to illustrate the type of data that the methodologies can yield. This submission is a protocol only, there is no data interpretation, and no conclusions are drawn on the findings. A copy of the related work has been attached to this submission.] Please clarify whether this conference proceeding or publication was peer-reviewed and formally published. If this work was previously peer-reviewed and published, in the cover letter please provide the reason that this work does not constitute dual publication and should be included in the current manuscript.

6. We note that Figures 1 and 7 in your submission contain copyrighted images. All PLOS content is published under the Creative Commons Attribution License (CC BY 4.0), which means that the manuscript, images, and Supporting Information files will be freely available online, and any third party is permitted to access, download, copy, distribute, and use these materials in any way, even commercially, with proper attribution. For more information, see our copyright guidelines: http://journals.plos.org/plosone/s/licenses-and-copyright.

a. You may seek permission from the original copyright holder of Figures 1 and 7 to publish the content specifically under the CC BY 4.0 license. 

Reviewers' comments:

Reviewer's Responses to Questions

**Comments to the Author**

1. Is the manuscript technically sound, and do the data support the conclusions?

Reviewer #1: Yes

Reviewer #2: Yes

Reviewer #3: Partly

2. Has the statistical analysis been performed appropriately and rigorously? 

Reviewer #1: Yes

Reviewer #2: Yes

Reviewer #3: Yes

3. Have the authors made all data underlying the findings in their manuscript fully available?

Reviewer #1: Yes

Reviewer #2: Yes

Reviewer #3: No

4. Is the manuscript presented in an intelligible fashion and written in standard English?

Reviewer #1: Yes

Reviewer #2: Yes

Reviewer #3: Yes

5. Review Comments to the Author

Reviewer #1: The paper is very important for everyone who works with different types of animal challenges. In the article under analysis, the proposed objective is to evaluate the different types of challenge (intranasal and aerosol). However, I would like to point out some comments and suggestions:

1- In the methodology, it is highlighted that intraperitoneal anesthesia is used intraperitoneally to perform the intranasal challenge, but this procedure does not happen when the aerosol challenge is performed. This anesthesia procedure causes stress to the animals, as well as alters the respiratory flow thus facilitating the administration of the bacterium. In addition, the intraperitoneal anesthesia procedure can develop inflammatory processes during injection, thus stimulating the response of inflammatory cells as well as the production of cytokines and chemokines. Therefore, it would be necessary to have the experimental groups with animals without anesthesia (naïve) and the group that had received anesthesia and evaluated the inflammatory process triggered from intraperitoneal anesthesia in animals challenged with aerosol.

2- In the aerosol challenge methodology, it is necessary to highlight more clearly whether it is performed individually or in groups of animals, and the size of the box that is used, because these factors would be limiting for the concentration of bacteria used, as well as the time that the animal should remain under aerosol.

3- Regarding the presentation of the graphs, it is recommended that the values of the scales be standardized, so that it is possible to compare the results and the groups.

4- Given the proposed objective and knowing the parameters to evaluate the challenge in the sense of even comparing the effectiveness of each type of test, it would be important to evaluate the weight of the animals, as well as the clinical score after the challenge. These parameters are important for evaluating the functionality and health of animal challenge outcomes.

Finally, I would like to congratulate the drawings, especially the experimental design, which made the objective and understanding of the article even clearer.

Reviewer #2: I found the article interesting, with effective experiments to prove the proposed question, with a clear objective, and the conclusion of the article correctly answers the question raised. I recommend publishing the article.

Reviewer #3: It is an interesting work and can increase what we know about the B. pertussis Challenge. I think the statistic were well conducted as it seems that the authors addressed correctly the statistics test.

But I missed one information that for me is crucial to analyses the results and also the conclusion.

The authors describe in the methods two strains of B. Pertussis UT25Sm1 and D420. For me is not clear where they use one or another. If use both in the both assays, is also not clear in the results which result is from UT25Sm1 and D420. In the results and also in the figure 2 which strain were used to generated these results? Please can you clarify and indicates in the methods, in the results and also in the figures which one were used?

As I it is no clear about two strains used, maybe were used one for each kind of experiment, or for each assay, if there are some differences in strain used in each assay, I could not support the conclusion as it can be different regarding the strain used. Even the expression of Bvg+ and hemolytic and colony morphology were the same, one strain can replicate quickly than other, or can lead a high immune response leading to the leukocytosis and pro-inflammatory cytokines differences that cannot be necessarily linked with the rout of administration So, it is crucial for me to identify why you have different strains and where you used each one. This can chance all the way we looked to the conclusions.

So, without this clarification I could not move on in the deep analysis of the conclusions linked with the results. I had marked as minor revision but depend on the clarification of this information it can be major revision. As the authors needs to compare more about two strains like the differences in the virulence factors, also the differences of the reactogenicity of each one that can lead to differences in the response, can be like performing one assay side by side with the two strains and so one...lets clarify first the missing information.

6. PLOS authors have the option to publish the peer review history of their article (what does this mean?). If published, this will include your full peer review and any attached files.

Reviewer #1: No

Reviewer #2: No

Reviewer #3: No

---

## [Author Response · Author response to Decision Letter 0]

20 Mar 2023

We would like to thank the reviewers for taking the time to review this manuscript and provide valuable insights. We have modified the figures to now include a new Figure S1 that depicts the wet lung weight of the mice 3 and 7 days post-challenge. We have also included the cytokine data for the mice challenged with D420 as Figure S6. The text has also been edited in response to reviewer’s comments.

Reviewer #1: The paper is very important for everyone who works with different types of animal challenges. In the article under analysis, the proposed objective is to evaluate the different types of challenge (intranasal and aerosol). However, I would like to point out some comments and suggestions:

1- In the methodology, it is highlighted that intraperitoneal anesthesia is used intraperitoneally to perform the intranasal challenge, but this procedure does not happen when the aerosol challenge is performed. This anesthesia procedure causes stress to the animals, as well as alters the respiratory flow thus facilitating the administration of the bacterium. In addition, the intraperitoneal anesthesia procedure can develop inflammatory processes during injection, thus stimulating the response of inflammatory cells as well as the production of cytokines and chemokines. Therefore, it would be necessary to have the experimental groups with animals without anesthesia (naïve) and the group that had received anesthesia and evaluated the inflammatory process triggered from intraperitoneal anesthesia in animals challenged with aerosol.

We agree with this reviewer that anesthesia could play a role in the inflammatory process and response to infection between both models. We have added text lines 450-452 and 454-456 to further discuss this caveat and added appropriate references. As suggested by this reviewer, one possible control would be animals without anesthesia. It is unfortunately not feasible to compare mice challenged with B. pertussis intranasally with vs without anesthesia as this intranasal administration cannot be performed on non-anesthetized animals (i.e. difficulty in handling, dose swallowing or rejection, animal discomfort). IACUC committees would also be unlikely to approve this type of experiment due to animal welfare concerns. The other control would be to compare mice that receive B. pertussis via aerosol to mice that are first anesthetized and then infected via aerosol. While this would be the best comparison regarding the inflammatory response generated by the anesthetic, we anticipate that the amount of bacteria delivered would vary as breathing rate and volume significantly decreases in anesthetized mice compared to alert mice. In addition, this experimental group would not be representative of any other methodology used in the field. 

2- In the aerosol challenge methodology, it is necessary to highlight more clearly whether it is performed individually or in groups of animals, and the size of the box that is used, because these factors would be limiting for the concentration of bacteria used, as well as the time that the animal should remain under aerosol.

Additional information has been added in the method section lines 124 and 125.

3- Regarding the presentation of the graphs, it is recommended that the values of the scales be standardized, so that it is possible to compare the results and the groups.

Thank you for pointing this out. We have modified Figure 2 panel D to match panel E, and have modified all the panels in Figure 3 to standardize the scales. We also modified panel B of Figure 5 to match panel A, Figure S1 ABDE to match CF and Figure S2B to match S2A.

4- Given the proposed objective and knowing the parameters to evaluate the challenge in the sense of even comparing the effectiveness of each type of test, it would be important to evaluate the weight of the animals, as well as the clinical score after the challenge. These parameters are important for evaluating the functionality and health of animal challenge outcomes.

We agree that weight of animal and clinical score after challenge can be used to monitor challenge. However, in these models, mice do not significantly display weight loss or changes in clinical scores over the period of 7 days, even with the highest challenge doses. For this reason, neither body weight nor clinical scores are shown.

We have now added new data in Figure S1. The new data depicts the change in wet lung weights at days 3 and 7 post-challenge in IN and aerosol-challenged mice. While we did not observe statistical differences between the groups compared throughout the study, mice challenged with B. pertussis IN tend to have higher wet lung weights, in particular at day 3 post-challenge. The text on lines 324-330 has been modified to describe these observations.

Finally, I would like to congratulate the drawings, especially the experimental design, which made the objective and understanding of the article even clearer.

Thank you.

Reviewer #2: I found the article interesting, with effective experiments to prove the proposed question, with a clear objective, and the conclusion of the article correctly answers the question raised. I recommend publishing the article.

Thank you for taking the time to review this manuscript.

Reviewer #3: It is an interesting work and can increase what we know about the B. pertussis Challenge. I think the statistic were well conducted as it seems that the authors addressed correctly the statistics test.

But I missed one information that for me is crucial to analyses the results and also the conclusion.

The authors describe in the methods two strains of B. Pertussis UT25Sm1 and D420. For me is not clear where they use one or another. If use both in the both assays, is also not clear in the results which result is from UT25Sm1 and D420. In the results and also in the figure 2 which strain were used to generated these results? Please can you clarify and indicates in the methods, in the results and also in the figures which one were used?

As I it is no clear about two strains used, maybe were used one for each kind of experiment, or for each assay, if there are some differences in strain used in each assay, I could not support the conclusion as it can be different regarding the strain used. Even the expression of Bvg+ and hemolytic and colony morphology were the same, one strain can replicate quickly than other, or can lead a high immune response leading to the leukocytosis and pro-inflammatory cytokines differences that cannot be necessarily linked with the rout of administration So, it is crucial for me to identify why you have different strains and where you used each one. This can chance all the way we looked to the conclusions.

So, without this clarification I could not move on in the deep analysis of the conclusions linked with the results. I had marked as minor revision but depend on the clarification of this information it can be major revision. As the authors needs to compare more about two strains like the differences in the virulence factors, also the differences of the reactogenicity of each one that can lead to differences in the response, can be like performing one assay side by side with the two strains and so one...lets clarify first the missing information.

In these studies, both UT25 and D420 were cultured on BG agar and SSM broth prior to administration. We have extensive experience with both of these strains as well as 30 additional genome sequenced representative isolates from around the US. We directly collaborate with the CDC on theses projects and we performed: RNAseq, toxin analysis, proteomics, naïve challenged, vaccinated, and in vivo RNA seq analysis with these strains. Please watch for several extensive papers that Mr. Graham Bitzer will be submitting for publication in the coming months. Our work on these strains led us to evaluate administration routes of nasal in comparison to aerosol. These studies were designed to answer simple questions about what route is most appropriate for challenge studies. Based on all of our data, we can assure the reviewer that these strains were virulent and hemolytic at the time of challenge. 

The strain used for each experiment is listed in both the method section (lines 133-137) and in the figure legend for each figure. For coherence, all the experiments shown in Figures 2 to 6 were performed with UT25. We used this strain as it is broadly used un the field to study pathogenesis of B. pertussis and vaccine efficacy in mice by both our lab and others. In addition, we repeated all the experiments with strain D420, which has been used extensively in the green olive baboon preclinical model of pertussis by the Tod Merkel Lab (FDA). Furthermore, D420 is a part of CDC clade013 and when isolated in 2012 the clade represented over 50% of recovered isolates in the US. D420 is currently the only strain used in baboons and it is going to be used in future human challenge studies that are being developed. 

Overall, we did not observe differences in how both strains behaved in the experimental models tested. For transparency, the data generated with D420 is included in Figures S5 and S6.

---

## [Decision Letter · Decision Letter 1]

26 May 2023

Intranasal challenge with B. pertussis leads to more severe disease manifestations in mice than aerosol challenge

PONE-D-22-33512R1

Dear Dr. Barbier,

We’re pleased to inform you that your manuscript has been judged scientifically suitable for publication and will be formally accepted for publication once it meets all outstanding technical requirements.

Kind regards,

Paulo Lee Ho, Ph.D.

Academic Editor

PLOS ONE

Additional Editor Comments (optional):

Reviewers' comments:

Reviewer's Responses to Questions

**Comments to the Author**

1. If the authors have adequately addressed your comments raised in a previous round of review and you feel that this manuscript is now acceptable for publication, you may indicate that here to bypass the “Comments to the Author” section, enter your conflict of interest statement in the “Confidential to Editor” section, and submit your "Accept" recommendation.

Reviewer #1: All comments have been addressed

Reviewer #3: All comments have been addressed

2. Is the manuscript technically sound, and do the data support the conclusions?

Reviewer #1: Yes

Reviewer #3: Yes

3. Has the statistical analysis been performed appropriately and rigorously? 

Reviewer #1: Yes

Reviewer #3: Yes

4. Have the authors made all data underlying the findings in their manuscript fully available?

Reviewer #1: Yes

Reviewer #3: Yes

5. Is the manuscript presented in an intelligible fashion and written in standard English?

Reviewer #1: Yes

Reviewer #3: Yes

6. Review Comments to the Author

Reviewer #1: All requested comments were answered in an integral and objective manner, showing the requested results and explanations.

Reviewer #3: The authors clarify the missing information’s about the uses of the 2 strains (UT25Sm1 and D420) and put on the missing information in the text as well the legend of the figure which one of the strains they use, and also insert the figure of the experiment with the second strain as well as which one the results are. With this we can see each result and not just believe in it, to extrapolate the results and the increase external validity of the experiment.

7. PLOS authors have the option to publish the peer review history of their article (what does this mean?). If published, this will include your full peer review and any attached files.

Reviewer #1: No

Reviewer #3: No

---

## [Editor Report · Acceptance letter]

7 Jun 2023

PONE-D-22-33512R1 

Intranasal challenge with *B. pertussis* leads to more severe disease manifestations in mice than aerosol challenge 

Dear Dr. Barbier:

I'm pleased to inform you that your manuscript has been deemed suitable for publication in PLOS ONE. Congratulations! Your manuscript is now with our production department. 

Kind regards, 

on behalf of

Dr. Paulo Lee Ho 

Academic Editor

PLOS ONE